# The Smoluchowski Ensemble—Statistical Mechanics of Aggregation

**DOI:** 10.3390/e22101181

**Published:** 2020-10-20

**Authors:** Themis Matsoukas

**Affiliations:** Department of Chemical Engineering, Pennsylvania State University, University Park, PA 16802, USA; txm11@psu.edu

**Keywords:** statistical thermodynamics, irreversible aggregation, Smoluchowski equation, gelation, phase transitions

## Abstract

We present a rigorous thermodynamic treatment of irreversible binary aggregation. We construct the Smoluchowski ensemble as the set of discrete finite distributions that are reached in fixed number of merging events and define a probability measure on this ensemble, such that the mean distribution in the mean-field approximation is governed by the Smoluchowski equation. In the scaling limit this ensemble gives rise to a set of relationships identical to those of familiar statistical thermodynamics. The central element of the thermodynamic treatment is the selection functional, a functional of feasible distributions that connects the probability of distribution to the details of the aggregation model. We obtain scaling expressions for general kernels and closed-form results for the special case of the constant, sum and product kernel. We study the stability of the most probable distribution, provide criteria for the sol-gel transition and obtain the distribution in the post-gel region by simple thermodynamic arguments.

## 1. Introduction

Aggregation is the process of forming structures through the merging of clusters. This generic process is encountered in a large variety of systems, from polymerization and colloidal aggregation to the clustering of social groups and the merging of galaxies. The mathematical foundations of aggregation were set by Smoluchowski [1], whose particular interest was in Brownian coagulation. The aggregation equation, more commonly known as Smoluchowski equation, is a rate equation on a distribution of clusters whose size (mass) changes by binary aggregation events. For a discrete population of clusters with integer masses in multiples of a unit mass (“monomer”) it takes the form [1],
(1)$$\frac{dc_{k}}{dt}=\frac{1}{2}\sum_{j=1}^{k-1}c_{k-j}c_{j}K_{k-j,j}-\sum_{j=1}^{\infty }c_{k}c_{j}K_{k,j},$$
where $c_{k}$ is the number concentration of clusters with mass *k* and $K_{i,j}$ the aggregation kernel, a rate constant for the merging of masses *i* and *j*. A large body of literature has focused on the theory of the Smoluchowski equation, the existence of analytic solution and the scaling limit [2]. Of particular interest is *gelling*, a condition that arises under the product kernel $K_{i,j}=ij$; it refers to the formation of a giant structure, as in polymer gels, and is manifested by the failure of the Smoluchowski equation to conserve mass. This process is commonly described as a phase transition, suggesting the possibility that statistical thermodynamics, a theory developed for equilibrium states of interacting particles, may perhaps be applicable in this clearly irreversible process.

Studies of Smoluchowski aggregation broadly fall in one of two categories, kinetic and stochastic. The kinetic approach is based on Equation (Equation 1) and its solution. Stable solutions conserve mass; gelling is identified as the point where mass conservation breaks down [3,4]. Post-gel solutions require additional assumptions as to how the gel and the dispersed phase interact [5]. The stochastic approach views clusters as entities that merge with probabilities proportional to the aggregation kernel. It was first formulated by Marcus [6] for a discrete finite population, and its formal mathematical treatment was developed by Lushnikov, who obtained solutions for certain special cases, including gelation [7,8,9,10,11]. In Lushnikov’s method all feasible distributions are given a probability, whose evolution in time is tracked via a generating functional. The approach is explicitly probabilistic and views the Smoluchowski equation as the mean-field approximation of the underlying stochastic process [12]. A different approach within the probabilistic realm makes use of combinatorial methods. This treatment originated with Stockmayer [13] and was further explored by Spouge [14,15,16,17]. The combinatorial approach considers the number of ways to build a particular distribution of clusters and assigns probabilities in proportion to that combinatorial weight. The ensemble of distributions is then reduced to the most probable distribution, which is identified by maximizing the combinatorial weight. This approach has two appealing advantages. It deals with a time-free ensemble in which time appears implicitly via the mean cluster mass. More importantly, it brings the problem closer to the viewpoint of statistical mechanics and the notion that an ensemble may be represented in the scaling limit by its most probable element. Stockmayer recognized this connection and his treatment of gelation is replete with references to the theory of phase transitions [13]. The analogy between aggregation and thermodynamics was not formalized, however. Stockmayer obtained the gel point by mathematical, not thermodynamic methods, and arrived at a post-gel solution that is not consistent with the kinetics of aggregation [5].

We have previously shown that gelation can be indeed treated as a formal phase transition and have presented solutions for the product kernel in the pre- and post-gel regions [18] based on our earlier work on the cluster ensemble [19,20]. Here we generalize the methodology to formulate a rigorous thermodynamic theory of Smoluchowski aggregation. We begin with a finite population that starts from a well defined state and construct the set of all possible distributions that can be reached in a fixed number of elementary transitions. The probability of distribution in this ensemble is governed by the kinetics of the elementary processes that act on the population. In the thermodynamic limit the most probable distribution is overwhelmingly more probable than all others and is governed by a set of mathematical relationships that we recognize as *thermodynamics*. The work is organized as follows. In Section 2 we define the Smoluchowski ensemble of distributions and their probabilities. In Section 3 we formulate the probability of distribution in terms of a special functional *W* that introduces the partition function and the Shannon entropy of distribution. In Section 4 we treat the scaling limit and derive the thermodynamic relationships of the Smoluchowski ensemble. In Section 5 we obtain solutions of the Gibbs form for the classical kernels, constant, sum and product. We analyze the stability and phase behavior of the ensemble in Section 6 and treat the sol-gel process as a phase transition. In Section 7 we express the results in the continuous domain and finally offer concluding remarks in Section 8.

## 2. The Smoluchowski Ensemble

We consider a population of clusters composed of i=1,2⋯ units (monomers). In binary aggregation two clusters merge to form a new cluster that conserves mass, via the schematic reaction
(2)$$\left(i\right)+\left(j\right)\overset{K_{i,j}}{\to }\left(i+j\right).$$The merging of a pair constitutes an elementary stochastic event, whose probability depends on the aggregation kernel $K_{i,j}$. At the initial state the population consists of $N_{0}=M$ single members (monomers). This distribution constitutes generation g=0. The next generation is constructed by implementing every possible aggregation event in the distribution of generation g=0. The set of distributions formed in this manner constitutes the microcanonical ensemble of generation g=1. We continue recursively to form the ensemble of distributions in generation *g* by implementing all possible aggregation events, one at a time, in all distributions of the parent ensemble. We represent a distribution of clusters by the vector $n=(n_{1},n_{2}\cdots )$, where $n_{i}$ is the number of clusters with *i* members. All distributions in generation *g* satisfy the conditions
(3)$$\sum_{i}n_{i}=M-g=N,\;\sum_{i}in_{i}=M.$$The first condition expresses the fact each elementary event decreases the number of clusters by 1, according to the stoichiometry of binary merging; the second condition expresses the fact that the number of members is conserved. Conversely, any distribution that satisfies the conditions in Equation (Equation 3) is a member of the ensemble of generation *g* because it can be formed in *g* steps from *M* monomers. We view the two equations in Equation (Equation 3) as the constraints that define the ensemble of feasible distributions. We call this ensemble microcanonical to indicate that it is conditioned by two extensive constraints that fix the mean cluster mass $M/N=\bar{x}$ in all distributions of the ensemble.

The evolution of the ensemble may be represented in the form of a layered graph (Figure 1), whose vertices represent distributions and edges represent elementary transitions according to Equation (Equation 2). Edges are directed from parent in generation g−1 to offspring in generation *g*. Layers are organized by generation and contain all distributions in a generation. The graph begins in generation g=0 with a distribution of all monomers and ends when all units have joined the same cluster. Stochastic aggregation is a random walk on this graph. A trajectory is a possible sequence of connected edges from top to bottom. Our goal is to establish the probability P(n) of distribution in generation g=0,1,⋯M−1, in terms the aggregation kernel $K_{i,j}$ for any *M*.

### 2.1. Kinetics

When cluster masses i−j and *j*, in distribution $n^{\prime }$ of generation g−1, merge to form a cluster of mass *i*, the parent distribution $n^{\prime }$ is transformed to offspring distribution n via the transition
(4)$$n^{\prime }\overset{\left(i-j)+(j)\to (i\right)}{\to }n.$$This transition is represented by an edge in the graph of Figure 1. Its rate $R_{i-j,j}$ is proportional to the number of ways to choose the reactants and the proportionality factor is the aggregation kernel:(5)$$R_{i-j,j}=K_{i-j,j}\frac{n_{i-j}^{\prime }\left(n_{j}^{\prime }-\delta_{i-j,j}\right)}{1+\delta_{i-j,j}}.$$The total rate by which parent $n^{\prime }$ produces offspring is
(6)$$R\left(n^{\prime }\right)=K\left(n^{\prime }\right)\frac{N^{\prime }\left(N^{\prime }-1\right)}{2},$$
where $N^{\prime }=\sum_{i}n_{i}^{\prime }$ is the number of clusters and $K(n^{\prime })$ is the mean kernel in parent distribution $n^{\prime }$:(7)$$K\left(n^{\prime }\right)=\frac{2}{N^{\prime }\left(N^{\prime }-1\right)}\sum_{i}\sum_{j}K_{i,j}\frac{n_{i}^{\prime }\left(n_{j}^{\prime }-\delta_{i,j}\right)}{1+\delta_{i,j}}.$$

In physical terms the aggregation kernel $K_{i,j}$ is the rate constant for the reaction between masses *i* and *j*. Its mathematical form may be constructed on the basis of a kinetic model for the particular problem. It is beyond the scope of this work to review the numerous kernels that have been proposed in the literature. We mention a selected few that are important for their physical, mathematical and historical significance, and summarize them in Table 1.

The Brownian coagulation kernel was derived by Smoluchowski [1] to describe the kinetics of diffusion limited aggregation in colloidal systems. The constant kernel was adopted by Smoluchowski [1] as an approximation for the Brownian kernel, a simplification that allows analytic results. This kernel is obtained by setting i=j in the Brownian kernel. The Flory/Stockmayer kernel [13,21] is a model for polymerization of chains composed of monomers with *f* functional groups. Assuming no cycles, a polymer with *i* monomers contains fi−2i+2 unreacted functional groups that are available to react. The Flory/Stockmayer kernel is the product of the unreacted functional groups in the two chains that merge. This kernel leads to gelation [13]. The product kernel is the limiting form of the Flory/Stockmayer kernel when the number of functional groups approaches infinity. It also leads to gelation, and being a simpler kernel than the Flory/Stockmayer, it serves as the standard model to study gelation. The sum kernel is proportional to the number of units in each cluster. This kernel may be viewed as the limiting form of a Flory/Stockmayer type kernel with two kinds functional groups [15], but its significance is primarily mathematical as one of a handful of kernels that lead to analytic solutions.

We discuss the constant, sum and product kernel in detail in Section 5. For now we leave the kernel general and unspecified. We only place the minimum conditions, $K_{i,j}=K_{j,i}>0$, which are required from elementary physical considerations; additionally, we adopt the normalization $K_{1,1}=1$.

### 2.2. Probabilities

We assign a probability P(n) to each distribution n within generation *g* and formulate its propagation in proportion to the probability of the parent and the rate of the parent-offspring transition:(8)$$P\left(n\right)=\sum_{n^{\prime }}P\left(n^{\prime }\right)\frac{R_{i-j,j}}{{\langle R\rangle }_{g-1}}.$$Here n is a distribution in generation *g*, $n^{\prime }$ is its parent of n in generation g−1 via the reaction (i−j)+(j)→(i), $R_{i-j,j}$ is the rate of the reaction, and ${\langle R\rangle }_{g-1}$ is the mean reaction rate in parent generation g−1:(9)$${\langle R\rangle }_{g-1}=\sum_{n^{\prime }}P\left(n^{\prime }\right)R\left(n^{\prime }\right).$$In both Equations (Equation 8) and (Equation 9) the summations are over all distributions $n^{\prime }$ in generation g−1. Expressing the transition rate in terms of the aggregation kernel we obtain
(10)$$\frac{R_{i-j,j}}{{\langle R\rangle }_{g-1}}=\frac{2}{N^{\prime }\left(N^{\prime }-1\right)}\frac{K_{i-j,j}}{{\langle K\rangle }_{g-1}}\frac{n_{i-j}^{\prime }\left(n_{j}^{\prime }-\delta_{i-j,j}\right)}{1+\delta_{i-j,j}}.$$
with
(11)$${\langle K\rangle }_{g-1}=\sum_{n^{\prime }}P\left(n^{\prime }\right)K\left(n^{\prime }\right).$$We may confirm that P(n) as defined in Equation (Equation 8) satisfies normalization over all distributions in the same generation. Beginning with $P(n_{0})=1$ at the initial state, Equation (Equation 8) uniquely determines the probabilities of all distributions in all future generations.

### 2.3. Smoluchowski Equation

The mean number of clusters with mass *k* in generation *g* is
(12)$$\langle n_{k}\rangle =\sum_{n}n_{k}P\left(n\right),$$
with the summation going over all distributions in the same generation. We will derive the evolution of the mean distribution from parent generation g−1 to generation *g*. The probability of distribution P(n) is given by Equation (Equation 8) and is expressed as a summation over its parents $n^{\prime }$. By the stoichiometry of the transition in Equation (Equation 2), the parent and offspring distributions satisfy
(13)$$n_{k}=n_{k}^{\prime }+\delta_{k,i}-\delta_{k,i-j}-\delta_{k,j}.$$Combining this relationship with (Equation 12) and (Equation 8) the result is (see Appendix A)
(14)$$\begin{array}{c} \langle n_{k}\rangle -\langle n_{k}^{\prime }\rangle =\frac{2}{N\left(N+1\right){\langle K\rangle }_{M,N+1}}{\langle \frac{1}{2}\sum_{j=1}^{k-1}n_{k-j}\left(n_{j}-\delta_{k-j,j}\right)K_{k-j,j}-\sum_{j=1}^{\infty }n_{k}\left(n_{j}-\delta_{k,j}\right)K_{k,j}\rangle }_{M,N+1}. \end{array}$$
The left-hand side is the change in the mean number of *k*-mers between generations; the right-hand side is the ensemble average of the production and depletion of *k*-mers within all distributions of the parent ensemble. Define the mean time increment Δt from parent to offspring generation as
(15)$$\Delta t{|}_{g-1\to g}=\frac{2}{N\left(N+1\right){\langle K\rangle }_{M,N+1}};$$
then Equation (Equation 14) reads
(16)$${\left.\frac{\Delta \langle n_{k}\rangle }{\Delta t}\right|}_{g-1\to g}={\langle \frac{1}{2}\sum_{j=1}^{k-1}n_{k-j}\left(n_{j}-\delta_{k-j,j}\right)K_{k-j,j}-\sum_{j=1}^{\infty }n_{k}\left(n_{j}-\delta_{k,j}\right)K_{k,j}\rangle }_{M,N+1}.$$
In the mean-field approximation we reduce the ensemble into a single distribution, $n^{*}$. This resolves the ensemble averages trivially and leads to the governing equation for $n^{*}$:(17)$$\frac{\Delta n_{k}^{*}}{\Delta t}=\frac{1}{2}\sum_{j=1}^{k-1}n_{k-j}^{*}\left(n_{j}^{*}-\delta_{k-j,j}\right)K_{k-j,j}-\sum_{j=1}^{\infty }n_{k}^{*}\left(n_{j}^{*}-\delta_{k,j}\right)K_{k,j}.$$
This is the Smoluchowski equation for binary aggregation, the discrete finite equivalent of Equation (Equation 1). The mean field approximation, which is invoked to obtain (Equation 17), implies that a single distribution is representative of the entire ensemble. In Section 4 and Section 6 we examine the conditions under which this is true.

## 3. Thermodynamic Formalism

### 3.1. Partition Function and Selection Functional

We now formulate the probability of distribution in terms of a special functional, W(n). It is through this formulation that we will make contact with statistical thermodynamics. We begin by writing the probability P(n) in generation *g* in the form
(18)$$P\left(n\right)=\frac{n!W(n)}{\Omega_{M,N}},$$
where n! is the multinomial coefficient of vector n,
(19)$$n!=\frac{(n_{1}+n_{2}\cdots )!}{n_{1}!n_{2}!\cdots }=\frac{N!}{n_{1}!n_{2}!\cdots },$$
N=M−g is the number of clusters in all distributions of generation *g*, W(n) is a functional of distribution n, to be determined, and $\Omega_{M,N}$ is the partition function. By the normalization condition on P(n) the partition function satisfies
(20)$$\Omega_{M,N}=\sum_{n}n!W\left(n\right),$$
with the summation over all distributions in generation g=M−N.

### 3.2. Shannon Entropy

The multinomial coefficient represents the combinatorial multiplicity of distribution n, namely, the number of ways to order the clusters in the distribution, if clusters with the same number of units are treated as indistinguishable. In the Stirling approximation, logx!=xlogx+O(logx), the log of the multinomial coefficient is
(21)$$\log n!=-\sum_{i}n_{i}\log\frac{n_{i}}{N}≐H\left(n\right).$$
It is a concave functional of n with functional derivatives
(22)$$\frac{\partial H(n)}{\partial n_{i}}=-\log\frac{n_{i}}{N}.$$
It is also homogeneous in n with degree 1 and satisfies the Euler condition
(23)$$H\left(n\right)=\sum n_{i}\frac{\partial H(n)}{\partial n_{i}}.$$
Setting $p_{i}=n_{i}/N$ and applying *H* to vector p we obtain
(24)$$H\left(p\right)=-\sum_{i}p_{i}\log p_{i}.$$
In this form *H* reverts to the familiar entropy functional, historically associated with Boltzmann, Gibbs and Shannon. We will call it *Shannon functional* and avoid the generic term “entropy,” whose meaning across disciplines varies. For our purposes the Shannon functional is defined as
(25)$$H\left(a\right)=H\left(a_{1},a_{2}\cdots \right)=-\sum_{i}a_{i}\log\frac{a_{i}}{\sum_{k}a_{k}}$$
and may be applied to any vector a with non-negative elements regardless of normalization.

### 3.3. The Selection Functional

Functional W(n) biases the statistical weight of distribution n relative to its combinatorial multiplicity. We call it selection functional because it effectively selects distributions relative to each other. The functional derivative of logW is
(26)$$\log w_{i;n}=\frac{\partial \log W(n)}{\partial n_{i}},$$
and defines the cluster function $w_{i;n}$, a property cluster mass *i* in distribution n. The cluster function $w_{i;n}$ depends not only on *i* but also on the distribution n on which this factor is evaluated. In the special case that logW is linear functional of n the functional derivative is a function of *i* alone and is the same in all distributions. This special condition is associated with Gibbs distributions, which are discussed in Section 5.

If W(n)=1 for all distributions, then the probability of distribution is proportional to its combinatorial multiplicity n!. If this special condition is met we will call the ensemble *unbiased*. The partition function of the unbiased ensemble can be easily determined by a combinatorial argument: it is equal to number of ways to assign *M* objects into *N* groups and is given by [22]
(27)ΩM,N∘=M−1N−1.
Accordingly, the probability of distribution in this special case is
(28)P∘(n)=n!M−1N−1.
In a population undergoing transformations, for example aggregation, fragmentation and so forth, the selection functional is determined by the kinetic details of the mechanisms that produce these transformations; in the case of aggregation it is determined by the aggregation kernel $K_{i,j}$. The question arises whether the unbiased ensemble is a possible solution of the Smoluchowski ensemble under some kernel. The answer is yes, and is given in Section 5.

### 3.4. Propagation Equations

At the initial state all clusters are monomers and the distribution is $n_{i,0}=M\delta_{i,1}$. We set $W(n^{0})=1$ and since $n^{0}!=1$ we also have $\Omega_{M,M}=1$. We insert Equation (Equation 18) into Equation (Equation 8) and express the summation over parents of n as a summation over all pairs (i−j,j) that produce mass *i* in distribution n. The result is (see Appendix A)
(29)$$\frac{\Omega_{M,N+1}}{\Omega_{M,N}}=\left(\frac{M-N}{N}\frac{1}{{\langle K\rangle }_{M,N+1}}\right)\left(\sum_{i=2}^{\infty }\frac{n_{i}}{M-N}\sum_{j=1}^{i-1}K_{i-j,j}\frac{W(n^{\prime })}{W(n)}\right).$$
Here *N* is the number of clusters in distribution n of generation g=M−N, $n^{\prime }$ is the parent distribution via the transition (i−j)+(j)→(i) and ${\langle K\rangle }_{M,N+1}$ is the mean kernel in the parent generation $g^{\prime }=g-1$. The left-hand side of Equation (Equation 29) depends solely on *M* and *N* whereas the second term on the right-hand side contains functionals of distribution n. This term must be the same for all distributions n in the same generation in order to produce a result that is a function of *M* and *N* alone. From Equation (Equation 18) it is clear that *W* and $\Omega_{M,N}$ may be defined within a proportionality constant $\alpha_{M,N}$; as long as this constant is common for all distributions in a generation it has no effect on probabilities and may be chosen arbitrarily. We choose it to satisfy the following criterion: if W=constant for all distributions, we require this constant to be 1. The choice that satisfies this condition is to set the double summation in Equation (Equation 29) to 1. Equation (Equation 29) now splits into two separate recursions, one for the partition function,
(30)$$\frac{\Omega_{M,N+1}}{\Omega_{M,N}}=\frac{M-N}{N}\frac{1}{{\langle K\rangle }_{M,N+1}}$$
and one for the selection functional,
(31)$$\sum_{i=2}^{\infty }\frac{n_{i}}{M-N}\sum_{j=1}^{i-1}\frac{W(n^{\prime })}{W(n)}K_{i-j,j}=1.$$
The recursion for the partition function is readily solved to produce the partition function in generation g=M−N:(32)$$\Omega_{M,N}=\Omega_{M,N}^{\circ }\prod_{\gamma =0}^{M-N+1}{\langle K\rangle }_{M,M-\gamma }.$$
Accordingly, the partition function is equal to the unbiased partition function times the product of all mean kernels from generation 0 up to the parent generation g−1. We write the recursion for the selection functional in the form
(33)$$W\left(n\right)=\sum_{i=2}^{\infty }\frac{n_{i}}{M-N}\sum_{j=1}^{i-1}K_{i-j,j}W\left(n^{\prime }\right).$$
The result gives the selection functional of the offspring as a linear combination of selection functionals of all its parents. In principle this can be solved recursively for any distribution in any generation. For certain special cases the recursion can be solved in closed form. These are discussed in Section 5.

## 4. Scaling Limit

### 4.1. Most Probable Distribution

We define the scaling limit by the condition M,N→∞ at fixed $M/N=\bar{x}$. The expectation is that in this limit the intensive mean distribution $\langle n_{k}\rangle /N$ must converge to a limiting distribution ${\bar{p}}_{k}$ that is independent of *M* and *N* and depends only on $M/N=\bar{x}$:(34)$$\frac{\langle n_{k}\rangle }{N}\to {\bar{p}}_{k}.$$
We further anticipate that the probability of distribution P(n) becomes infinitely sharp around a single distribution, $n^{*}=Np^{*}$, such that $p_{k}^{*}$ is not merely the most probable distribution, it is overwhelmingly more probable than any other distribution in the ensemble. This further implies that the mean distribution and most probable distribution converge to each other:(35)$$\langle p_{k}\rangle \to p_{k}^{*}.$$
This convergence is an implicit requirement for the validity of the Smoluchowski equation: the mean-field approximation is exact if a single distribution is representative of the entire ensemble. This is possible only if P(n) peaks very sharply about the most probable distribution. When a single term dominates the summation that defines the partition function in Equation (Equation 20), the log of the sum converges to the log of the maximum term,
(36)∑XXlogΩM,N=H(n*)+logW(n*),
with $H\left(n^{*}\right)=\log n^{*}!$. As a further consequence of the intensive convergence in (Equation 34) we have the Euler relationship for logW:(37)∑XXlogW(n*)=∑ini*logwi*.
where $\log w_{i}^{*}=\log w_{i;n^{*}}$ is the functional derivative of $\log W(n^{*})$,
(38)$$\log w_{i}^{*}=\frac{\partial \log W(n^{*})}{\partial n_{i}^{*}}.$$
Equation (Equation 37) expresses the fact that logW is homogeneous functional of the MPD. This condition follows from Equation (Equation 36) and the homogeneity properties of $H(n^{*})$ and $\log\Omega_{M,N}$.

The most probable distribution (MPD) maximizes the probability in Equation (Equation 18) among all distributions that satisfy the constraints in Equation (Equation 3). By Lagrange maximization we obtain the MPD in the form
(39)$$\frac{n_{k}^{*}}{N}=w_{k}^{*}\frac{e^{-\beta i}}{q},$$
and *q* and β are parameters related to the Lagrange multipliers. We insert the MPD into Equation (Equation 36) to obtain
(40)∑XXlogΩM,N=βM+(logq)N.
This fundamental equation relates the partition function to the primary variables of the ensemble: the macroscopic variables (M,N) that define the ensemble, and the Lagrange multipliers (β,q) that appear in the MPD. The convergence of $n_{k}^{*}/N$ to intensive limit $p_{k}^{*}$ implies that β and *q* are intensive, that is, they are functions of $\bar{x}=M/N$ but not of *M* or *N* individually. This further implies that Equation (Equation 40) is homogeneous function of *M* and *N* with degree 1 and thus must satisfy Euler’s theorem:(41)$$\log\Omega_{M,N}=\left(\frac{\partial \log\Omega_{M,N}}{\partial M}\right)M+\left(\frac{\partial \log\Omega_{M,N}}{\partial N}\right)N.$$
Direct comparison with Equation (Equation 40) leads to: (42)$$\begin{array}{c} \beta ={\left(\frac{\partial \log\Omega_{M,N}}{\partial M}\right)}_{N}, \end{array}$$(43)$$\begin{array}{c} \log q={\left(\frac{\partial \log\Omega_{M,N}}{\partial N}\right)}_{M}. \end{array}$$
Thus the Lagrange multipliers that appear in the MPD are the partial derivatives of the partition function. Differentiation of Equation (Equation 40) with respect to all variables that appear on the right-hand side gives
(44)Mdβ+Ndlogq=0.
This is the Gibbs-Duhem equation associated with the Euler equation for $\log\Omega_{M,N}$ in Equation (Equation 40). It may be written as
(45)$$\bar{x}=-\frac{d\log q}{d\beta }.$$
In this form its expresses the relationship between β, *q* and $\bar{x}$.

The MPD maximizes the log of the microcanonical weight, H(n)+logW(n) and its maximum is $\log\Omega_{M,N}$. Therefore we have the inequality:(46)∑XXlogΩM,N≥H(n)+logW(n).
It is satisfied by all distributions n in the (M,N) ensemble with the equal sign only for $n=n^{*}$. This is the fundamental variational principle of the ensemble: it defines the MPD and generates all relationships of this section.

### 4.2. Thermodynamics

We recognize the equations of the previous section as those of familiar statistical thermodynamics. Equation (Equation 39) is the generalized canonical distribution, a member of the exponential family, whose parameters β and *q* are related to the microcanonical partition function via Equations (Equation 40), (Equation 42) and (Equation 43). We define the extensive form of the canonical partition function Q(β,N) via the Legendre transformation of logΩ:(47)$$\log Q=\log\Omega -M{\left(\frac{\partial \log\Omega }{\partial M}\right)}_{N}=N\log q,$$
and thus we recognize $q=Q^{1/N}$ as the intensive form of the canonical partition function.

The variational condition that produces the set of thermodynamic relationships is the inequality in Equation (Equation 46), which defines the MPD as the distribution that maximizes the microcanonical weight. Expressing $H(n^{*})$ and $\log W(n^{*})$ in terms of the Euler relationships (Equation 21) and (Equation 37), respectively, this inequality takes the form
(48)$$\frac{\log\Omega_{M,N}}{N}\geq -\sum_{i}p_{i}\log\frac{p_{i}}{w_{i}^{*}},$$
where $p_{i}=n_{i}/N$. The inequality is satisfied by all distributions $p_{i}$ with mean $\bar{x}=M/N$ and the equality applies only to $p_{i}=p_{i}^{*}$. With $w_{i}^{*}=1$ it reduces the second law: the log of the microcanonical partition function is equal to the Shannon entropy of the most probable distribution, and this is larger than the entropy of any other distribution with the same mean.

Table 2 summarizes these relationships. They are consequences of the maximization of the probability in Equation (Equation 18) and are independent of the details of aggregation. These details enter only through Equations (Equation 32) and (Equation 33), which express the partition function and the selection functional in terms of the aggregation kernel.

## 5. Gibbs Distributions

A special type of functional is of the form
(49)$$W\left(n\right)=\prod_{i}w_{i}^{n_{i}},$$
whose log is linear in n
(50)$$\log W\left(n\right)=\sum_{i}n_{i}\log w_{i}$$
with functional derivative $\log w_{i}$. Here $w_{i}$ is a function of *i* alone and does not depend on n. If the selection functional is given by Equation (Equation 49) the probability of distribution in Equation (Equation 18) takes the form
(51)$$P\left(n\right)=\frac{N!}{\Omega_{M,N}}\prod_{i}\frac{w_{i}^{n_{i}}}{n_{i}!}.$$
Probability distributions of this type are called Gibbs distributions [23] and are frequently encountered in stochastic processes [24]. Several important results can be obtained in analytic form. In particular, the mean distribution is [22]:(52)$$\frac{\langle n_{k}\rangle }{N}=w_{k}\frac{\Omega_{M-k,N-1}}{\Omega_{M,N}}.$$
The result is exact for all 1≤N≤M, 1≤k≤M−N+1.

We apply this selection functional of Equation (Equation 49) to the transition (i−j)+(j)→(i) that converts parent distribution $n^{\prime }$ to offspring n. By the stoichiometry of the transition we have
(53)$$\frac{W(n^{\prime })}{W(n)}=\frac{w_{i-j}w_{j}}{w_{i}}.$$
Inserting into Equation (Equation 31) we obtain
(54)$$\sum_{i=2}^{\infty }\frac{n_{i}}{M-N}\sum_{j=1}^{i-1}\frac{w_{i-j}w_{j}}{w_{i}}K_{i-j,j}=1.$$
One possible solution that satisfies this equation for all distributions n is
(55)$$w_{i}=\frac{1}{i-1}\sum_{j=1}^{i-1}w_{i-j}w_{j}K_{i-j,j};\;w_{1}=1.$$
This is not the only possible solution for *W* in Equation (Equation 31) and may or may not be acceptable; if it is, we have obtained a Gibbs distribution and the kernel is a Gibbs kernel.

We have identified three kernels for which Equation (Equation 55) is the correct solution. These are the constant kernel,
(56)$$K_{i,j}=1,$$
the sum kernel
(57)$$K_{i,j}=\frac{i+j}{2}$$
and their linear combinations. The product kernel is a *quasi-Gibbs* kernel and is discussed in Section 5.3.

Here we provide detailed solutions for the constant and sum kernels. We will not discuss the linear combination in part because the results are more involved but mainly because this kernel reverts to the sum kernel when cluster masses are large thus it does not contribute to our understanding of aggregation beyond what we learn by studying the constant and sum kernels separately.

### 5.1. Constant Kernel

With $K_{i,j}=1$ Equation (Equation 55) gives $w_{i}=1$ for all *i*. Accordingly, the ensemble is *unbiased* and its partition function is given by Equation (Equation 27):(58)ΩM,N=ΩM,N∘=M−1N−1.
The mean distribution follows from Equation (Equation 52) and is given by
(59)nkN=M−k−1N−2M−1N−1.
To obtain the most probable distribution we calculate the parameters β and *q* from Equations (Equation 42) and (Equation 43) along with (Equation 58). The differentiations may be done by first replacing the factorials in the partition function with the Stirling expression. Alternatively we may obtain these parameters by the discrete difference form of these derivatives and apply the asymptotic conditions M,N≫1. The latter method is simpler:(60)$$\begin{array}{c} \beta =\log\frac{\Omega_{M+1,N}}{\Omega_{M,N}}=\frac{M}{M-N+1}\to \frac{\bar{x}}{\bar{x}-1}, \end{array}$$(61)$$\begin{array}{c} q=\frac{\Omega_{M,N+1}}{\Omega_{M,N}}=\frac{M-N}{N}=\bar{x}-1. \end{array}$$
We obtain the MPD from (Equation 39) with $w_{k}^{*}=1$:(62)$$\frac{n_{k}^{*}}{k}=\frac{1}{\bar{x}-1}{\left(\frac{\bar{x}}{\bar{x}-1}\right)}^{-k}.$$
For large $\bar{x}$ this goes over to the exponential distribution
(63)$$f\left(x\right)=\frac{e^{-x/\bar{x}}}{\bar{x}},$$
which is the well known result for the constant kernel. Here *x* stands for the continuous cluster mass.

### 5.2. Sum Kernel

The ensemble average of the sum kernel is
(64)$${\langle K\rangle }_{M,N}=\frac{M}{N}.$$
We obtain the partition function from Equation (Equation 32). The result is
(65)ΩM,N=N!MM−NM!M−1N−1.
The factors $w_{k}$ that satisfy Equation (Equation 55) are
(66)$$w_{k}=\frac{k^{k-1}}{k!}$$
and the mean distribution follows from (Equation 52),
(67)$$\frac{\langle n_{k}\rangle }{N}=\frac{k^{k-1}}{k!}\frac{{\left(M-k\right)}^{M-N-k}}{M^{M-N-1}}\frac{(N-1)(M-N)!}{N(M-N-k+1)!}.$$
This is an exact result for all 1≤N≤M, 1≤k≤M−N+1. The parameters β and *q* are obtained similarly to those for the constant kernel:(68)$$\begin{array}{c} \beta =\frac{\Omega_{M+1,N}}{\Omega_{M,N}}\to \frac{M-N}{M}-\log\frac{M-N}{M}, \end{array}$$(69)$$\begin{array}{c} q=\frac{\Omega_{M,N+1}}{\Omega_{M,N}}\to \frac{M-N}{M}. \end{array}$$
Combining with Equation (Equation 39) we obtain the MPD in the form
(70)$$\frac{n_{k}^{*}}{N}=\frac{k^{k-1}}{k!}\theta^{k-1}e^{k\theta },$$
with $\theta =1-1/\bar{x}$. We use the Stirling formula for the factorial the MPD in the continuous limit takes the form
(71)$$f\left(x\right)=\frac{\theta^{x-1}}{\sqrt{2\pi }}\frac{e^{-x\theta }}{x^{3/2}}.$$
Figure 2 shows the MPD for $\bar{x}=10$ and the mean distribution from Equation (Equation 67) at fixed M/N=10 for various values of *M* and *N*. In the scaling limit the mean distribution converges to the MPD.

### 5.3. Quasi-Gibbs Kernels—The Product Kernel

We are able to obtain closed-form expressions for the partition function of the constant and sum kernels and heir linear combinations because they all satisfy the condition
(72)$${\langle K\rangle }_{M,N}=K\left(n\right)$$
for all n. This states that the mean kernel is the same in all distributions of the ensemble, therefore also equal to the ensemble average kernel. In this case the calculation of the ensemble average kernel is trivial, as it does not require knowledge of the probabilities n. The constant kernel, sum kernel and their linear combinations are the only kernels that satisfy (Equation 72) in the strictest sense, that is, for all n that satisfy the two constraints in (Equation 3). We refer to Equation (Equation 72) as the Gibbs condition because it generates Gibbs distributions. We may relax the requirement that *all* distributions obey the Gibbs condition with the milder requirement that it be obeyed by *most* distributions. This is the case of the produce kernel. The product kernel is defined
(73)$$K_{i,j}=ij,$$
and its mean within distribution n is
(74)$$K\left(n\right)=\frac{N}{N-1}\left({\langle i\rangle }^{2}-\frac{\langle i^{2}\rangle }{N}\right).$$
Here $\langle i\rangle =M/N$ and $\langle i^{2}\rangle$ are the normalized first moment and second moments of n, respectively. In the limit N→∞, M/N=fixed, this scales as
(75)$$K\left(n\right)∼{\langle i\rangle }^{2}={\left(\frac{M}{N}\right)}^{2},$$
in most distributions except those that contain clusters of the order *M*. (The largest cluster size in the ensemble is $k_{\max}=M-N+1$ and for M≫N it is of the order *M*.) According to Equation (Equation 75) the product kernel is a quasi-Gibbs kernel: it satisfies the Gibbs condition in Equation (Equation 72) asymptotically in most but not all feasible distributions. We proceed to obtain the Gibbs distribution of the product kernel and test its validity.

Inserting (Equation 75) into (Equation 32) we obtain the partition function:(76)ΩM,N=N!MM−NM!2M−1N−1.
We complete the solution by evaluating $w_{k}$ from Equation (Equation 55),
(77)$$w_{k}=\frac{2^{k-1}k^{k-2}}{k!}.$$
The mean distribution is obtained by inserting these results into Equation (Equation 52):(78)$$\frac{\langle n_{k}\rangle }{N}=\frac{2^{k-1}k^{k-3}\left(N-1\right)M!M^{1-2(M-N)}\left(M-N\right)!{\left(M-k\right)}^{2(M-N-k)}}{N^{2}\left(k-1\right)!\left(M-k-1\right)!\left(M-N-k+1\right)!}.$$
Unlike Equations (Equation 62) and (Equation 67) this result is not exact. This can be demonstrated numerically by the fact this distribution is not normalized to unity and its mean is not M/N for finite *M*, *N*; its approaches proper normalization in the asymptotic limit. This failure arises from the fact that Equation (Equation 52) requires a Gibbs probability distribution that strictly applies to all distributions of the (M,N) ensemble.

We obtain β and logq from Equations (Equation 42) and (Equation 43):(79)$$\begin{array}{c} \beta =\frac{M-N}{M}-2\log\frac{M-N}{M} \end{array}$$(80)$$\begin{array}{c} q=\frac{N(M-N)}{M^{2}}. \end{array}$$
Using $\theta =1-1/\bar{x}$ the MPD is
(81)$$\frac{n_{k}^{*}}{N}=\frac{{\left(2\theta k\right)}^{k-2}}{k!}\frac{2\theta }{1-\theta }e^{-2\theta k}$$
and in the continuous limit
(82)$$f\left(x\right)=\frac{2^{x}e^{x(1-2\theta )}\theta^{x-1}}{\sqrt{8\pi }\left(1-\theta \right)x^{5/2}}.$$
These results are summarized in Table 3 along with those for the constant and sum kernels.

The relationship between the mean and the most probable distribution of the product kernel is shown in Figure 3 for two values of the mean cluster, $\bar{x}=1.75$ and $\bar{x}=4$. At $\bar{x}=1.75$ the mean distribution calculated from Equation (Equation 78) is not exact but its moments asymptotically approach the correct values as *M* and *N* are increased at fixed M/N. At $\bar{x}=4$ the behavior is different. A peak develops at the long tail of the distribution. It is pushed to ever larger sizes but never vanishes. In this region the mean distribution from Equation (Equation 78) is not correct: its mean does not converge to $\bar{x}$ when *M* and *N* are increased, but to a value smaller than $\bar{x}$, which implies that mass conservation is not satisfied. This breakdown is manifestation of *gelation*, the emergence of an infinite cluster that is not captured by the mean field theory. The precise nature of the gel phase is discussed in the next section.

## 6. Phase Behavior

### 6.1. Stability

The fundamental inequality of the ensemble is Equation (Equation 46) that defines the most probable distribution. This condition implies that the microcanonical functional is concave and this in turn implies that $\log\Omega_{M,N}$ is a concave *function* of *M* and *N* and requires (see Appendix A)
(83)$$\frac{d\beta }{d\bar{x}}\leq 0\;\mathrm{or}\;\frac{d\log q}{d\bar{x}}\geq 0.$$
These equivalent conditions guarantee the existence of the MPD in the form of Equation (Equation 39). In thermodynamic language they ensure that the MPD represents a stable state. The parameters β and *q* of the constant, sum and product kernel are plotted in Figure 4a,b, respectively, as a function of the progress variable $\theta =1-1/\bar{x}$. According to Equation (Equation 83) stability requires β to be decreasing function of $\bar{x}$ and *q* increasing function of $\bar{x}$. The constant kernel is stable at all θ: $\beta_{\mathrm{const}}$ decreases and $q_{\mathrm{const}}$ increases monotonically over the entire range of θ. The sum kernel is also stable at all θ but reaches the limit of stability at θ=1 or $\bar{x}=\infty$. This kernel is borderline-stable: it is stable for all finite times and reaches instability at t=∞. The product kernel is stable up to $\bar{x}=0.5$ beyond which point both $\beta_{\mathrm{prod}}$ and $q_{\mathrm{prod}}$ violate the stability criteria.

To survey the stability landscape of aggregation we employ the power-law kernel,
(84)$$K_{i,j}={\left(ij\right)}^{\nu /2},$$
with arbitrary exponent ν≥0. This is a homogeneous kernel with degree ν. It reverts to the product kernel with ν=2 and to the constant kernel with ν=0. We treat this as a quasi-Gibbs kernel by analogy to the product kernel. We take the ensemble average power-law kernel to scale as
(85)$${\langle K\rangle }_{M,N}∼{\left(\frac{M}{N}\right)}^{\nu },$$
and obtain the parameters β and *q* as
(86)$$\beta =\nu \theta -\log\theta ,\;q=\theta {\left(1-\theta \right)}^{\nu -1}.$$
With ν=0 and ν=2 these revert, as expected, to the results for the constant and product kernels, respectively. Interestingly, with ν=1 we obtain the (β,q) parameters for the sum kernel. This behavior turns the power-law kernel into a useful tool, a homogeneous kernel that reproduces the correct (β,q) values of the constant, sum and product kernels, and which may be used to interpolate (and cautiously extrapolate) to other homogeneous kernels by varying the exponent ν.

The stability limit in power-law aggregation is reached at
(87)$$\theta^{*}=1/\nu .$$
Accordingly, the MPD is stable in $0\leq \theta \leq \theta^{*}$ and unstable in $\theta^{*}<\theta \leq 1$. The phase diagram is shown in Figure 4a,b with the stable region indicated by the shaded area. For ν≤1 the system is stable at all θ from 0 to 1. For ν=1 the limit of stability appears at θ=1, which is reached in infinite time. In practice the system is stable at all finite times. For ν>1 the stability limit is reached within finite time at the point where the mean size reaches the critical value
(88)$${\bar{x}}^{*}=\frac{1}{1-\theta^{*}}=\frac{\nu }{\nu -1}.$$
For ν=2 (product kernel) the limit of stability is reached at ${\bar{x}}^{*}=2$. We see from Figure 4 that both β and *q* reach the limit of stability simultaneously.

### 6.2. Phase Splitting—The Sol-Gel Transition

When the system crosses into the unstable region its state is no longer represented by the MPD but by a mixture of two phases, each with its own MPD. What are these phases? To answer this question we first observe that the elements of the ensemble are fundamentally *discrete* distributions; the apparent continuity in the scaling limit is a mathematical artifact, a great convenience, but not a fundamental quality of the ensemble. To understand therefore the nature of the gel phase we must consider the discrete finite system. Given a distribution of *M* particles partitioned into *N* clusters, the maximum cluster mass possible is $k_{\max}=M-N+1$ and is found in a single distribution of the ensemble, in which one cluster contains M−N+1 units and the remaining N−1 clusters contain one unit mass each. The region $\left(k_{\max}+1\right)/2<k\leq k_{\max}$ is special: it is either empty, or it contains a single cluster. It cannot contain more than one cluster because there is not enough mass to have two clusters that are both larger than $(k_{\max}+1)/2$. In the event that it does contain a cluster, its mass is of the order of $k_{\max}=M-N+1$, and in the asymptotic limit, of the order *M*. This means that the mass in the region $k>k_{\max}+1)/2$ is of the same order as that in $k<(k_{\max}+1)/2$. A cluster in $k>(k_{\max}+1)/2$ represents a *giant component*, a single element of the population that carries a finite fraction of the total mass contained in the distribution.

The set of distributions that do not contain a giant cluster constitute the *sol* phase; sol distributions satisfy the scaling form of the mean kernel in Equation (Equation 85) and the Gibbs condition in Equation (Equation 72). Distributions that contain a cluster in the gel region violate the Gibbs condition and will be treated as a mixture of a sol phase $\left(k\leq \left(k_{\max}+1\right)/2\right)$ and a gel phase ($k>(k_{\max}+1)/2$). Given an individual distribution n, a certain fraction of mass is contained in the sol region with the rest in the gel region. The ensemble averages of these fractions define, respectively, the sol fraction, $\phi_{\mathrm{sol}}$, and gel fraction, $\phi_{\mathrm{gel}}$, in the ensemble:(89)$$\phi_{\mathrm{sol}}=\frac{1}{M}\sum_{n}P\left(n\right)\sum_{k=1}^{k^{\prime }}n_{k};\;\phi_{\mathrm{gel}}=\frac{1}{M}\sum_{n}P\left(n\right)\sum_{k=k^{\prime }+1}^{k_{\max}}n_{k};\;\phi_{\mathrm{sol}}+\phi_{\mathrm{gel}}=1,$$
with $k^{\prime }=\left(k_{\max}+1\right)/2$. If P(n) is such that in the scaling limit $\phi_{\mathrm{gel}}\to 0$, the ensemble consists of a single phase, the *sol*, and is represented by the MPD in Equation (Equation 39). If $\phi_{\mathrm{gel}}>0$ the ensemble is represented by a mixture of the two phases. We will determine their distributions and construct the tie line between the two phases.

We suppose that the state at (M,N) consists of a sol phase with $M_{\mathrm{sol}}$, $N_{\mathrm{sol}}=N-1$, and a gel phase with $M_{\mathrm{gel}}=M-M_{\mathrm{sol}}$. The evolution of the sol phase is governed by Equation (Equation 30), which we now write as
(90)$${\left(\frac{\Omega_{M+1,N}}{\Omega_{M,N}}\right)}_{\mathrm{sol}}=q\left(\theta_{\mathrm{sol}}\right)=\theta_{\mathrm{sol}}{\left(1-\theta_{\mathrm{sol}}\right)}^{\nu -1}.$$
This must be satisfied by the sol phase at all times. In the pre-gel region the state is a single phase, sol, with $\theta_{\mathrm{sol}}=\theta =1-N/M$ and (β,q) parameters from Equation (Equation 86). In the post-gel region it is a mixture of two phases: a sol phase with mass $M_{\mathrm{sol}}$ and number of clusters $N_{\mathrm{sol}}=N-1$; and gel phase with mass $M_{\mathrm{gel}}=M-M_{\mathrm{sol}}$ found in a single cluster ($N_{\mathrm{gel}}=1$). The sol phase is determined from Equation (Equation 90) with $\theta_{\mathrm{sol}}=1-M_{\mathrm{sol}}/\left(N-1\right)$ and its β-*q* parameters are given by Equation (Equation 86) with $\theta =\theta_{\mathrm{sol}}$. The mass of the gel phase is then obtained from the conservation conditions $M_{\mathrm{gel}}=M-M_{\mathrm{sol}}$. These steps are summarized below.


**Pre-Gel Region**
$0\leq \theta <\theta^{*}$


The system consists of a sol phase and its MPD is
(91)$$\frac{n_{k}^{*}}{N}=w_{k}^{*}\frac{e^{-\beta (\theta )}}{q(\theta )}$$
with
(92)$$\beta =\nu \theta -\log\theta ,\;q=\theta {\left(1-\theta \right)}^{\nu -1},\;\theta =1-N/M.$$


**Post-Gel Region**
$\theta^{*}\leq \theta <1$


The system consists of a sol phase with mass fraction $\phi_{\mathrm{sol}}$ and a gel phase with fraction $\phi_{\mathrm{gel}}=1-\phi_{\mathrm{sol}}$.
Obtain $\theta_{\mathrm{sol}}$ by solving
(93)$$q\left(\theta_{\mathrm{sol}}\right)=q\left(\theta \right),\;\theta_{\mathrm{sol}}\leq \theta^{*}.$$
with q(θ) from Equation (Equation 86) and θ=1−N/M.Obtain $\phi_{\mathrm{sol}}$ and ${\bar{x}}_{\mathrm{sol}}$ from
(94)$$\phi_{\mathrm{sol}}=\frac{1-\theta }{1-\theta_{\mathrm{sol}}},\;{\bar{x}}_{\mathrm{sol}}=\frac{1}{1-\theta_{\mathrm{sol}}}.$$Obtain the gel fraction from mass balance:
(95)$$\phi_{\mathrm{gel}}=1-\phi_{\mathrm{sol}}=\frac{\theta -\theta_{\mathrm{sol}}}{1-\theta_{\mathrm{sol}}}.$$The mean size of the gel cluster is ${\bar{k}}_{\mathrm{gel}}=\phi_{\mathrm{gel}}M$, where *M* is the total mass in the system. In the scaling limit the gel fraction is 1 and the size of the gel cluster is *∞*.

The gel fraction and the mean cluster size for the product kernel (ν=2) are shown in Figure 5 as a function of θ. The gel fraction is zero up until the gel point ($\theta^{*}=0.5$) and increases according to Equation (Equation 95) once in the post-gel region. The mean cluster size increases in the pre-gel region but decreases in the post-gel region, as clusters in the sol are lost by reaction with the gel. At θ→1 (t→∞) all mass is found in the gel phase except for a single sol particle with unit mass. This is the infinite dilution limit of the sol phase, to borrow the terminology of solution thermodynamics.

The evolution of ${\bar{x}}_{\mathrm{sol}}$ past the gel point retraces its pre-gel history. This is a consequence of Equation (Equation 93), which resolves the sol phase in the two-phase region. The symmetry of q(θ) about $\theta^{*}=0.5$ in the case of the product kernel produces a correspondingly symmetric evolution of ${\bar{x}}_{\mathrm{sol}}$, as shown in Figure 5b. The dashed lines are Monte Carlo simulations with M=200 particles and are shown for comparison (the simulations are discussed in the next section). The deviation from theory near the gel point is due to finite size effects. In these simulations a relatively small number of particles was used to permit the collection of a large number of realizations within reasonable computational time. These simulations are discussed next.

### 6.3. Monte Carlo Simulations

We demonstrate the theory with simulations performed by the constant-*V* Monte Carlo method [25]. The method tracks a sample of clusters that undergo binary aggregation with probability proportional to the transition rate $R_{i,j}$ given in Equation (Equation 5). At each step the simulation box contains a sample of *N* clusters, with *N* decreasing from *M* to 1 as clusters merge. A pair of clusters are chosen at random and is combined into a single cluster according the following criterion: draw a random number rnd in the interval (0,1) and accept the merging of the clusters if
(96)$$\mathrm{rnd}<\frac{K_{i,j}}{K_{\max}},$$
where $K_{i,j}$ is the aggregation kernel between the chosen clusters and $K_{\max}$ is the maximum aggregation kernel in the simulation box. If the criterion is satisfied the event is accepted and the reactant particles are deleted and replaced by a cluster with their combined mass. If the event is rejected, a new pair is chosen and the process is repeated. The simulation begins with *M* monomers and continues until a single cluster is formed. This amounts to a random walk along the edges of the graph in Figure 1 that spans its entire range from θ=0 to θ=1−1/M. A trajectory from the top to the bottom of the graph consists of a sequence of *M* sampled distributions, one from each generation. By averaging trajectories we obtain the mean distribution in each generation, which may then be compared to the mean distribution predicted by the theory.

Figure 6 shows the evolution of the mean distribution obtained by MC simulation with the product kernel using M=200. Up until the gel point is reached the state is a single sol phase. It is characterized by a population of clusters whose tail decays fast enough that its moments are finite. Above the gel point a gel peak emerges. It becomes more pronounced and moves to larger sizes as aggregation progresses. Past the gel point the sol distribution contracts and retraces its steps back to the monomeric state as θ increases. For example, the sol distribution at θ=0.9 is identical to that at θ=0.1 except that it carries less mass. In the Smoluchowski literature this is known as the Flory solution to gelation [5]. A competing solution by Stockmayer [13] predicts that the intensive distribution of the sol phase remains constant past the gel point except for the fact that its mass gradually decreases as it is transferred to the gel. As it turns out, Stockmayer solution implicitly assumes that P(n) is strictly a Gibbs distribution. In this case the sol-gel tie line is obtained by equating the temperatures of the two phases and the sol distribution is indeed found to be constant throughout the post gel region [20]. An analysis of the Stockmayer solution is beyond the scope of this work but a commentary is given in Reference [22].

## 7. Continuous Limit

We define the continuous limit by the conditions
(M,N)→∞,M/N≫1.
Thus in addition to the scaling limit we require the mean cluster size to be much larger than the unit mass, such that the cluster mass may be treated as a continuous variable, which we denote as *x*. Equations (Equation 63), (Equation 71) and (Equation 82) refer to this limit. We present the corresponding expressions for the partition function and the selection functional.

In the continuous domain all intensive properties of the ensemble are functions of the mean cluster size $\bar{x}$. Thus we write $\beta =\beta (\bar{x})$, $q=q(\bar{x})$, $w_{i}^{*}=w\left(x;\bar{x}\right)$, and express the partition function in intensive form $\log\omega \left(\bar{x}\right)=\left(\log\Omega_{M,N}\right)/N$. The MPD is
(97)$$f\left(x\right)=w\left(x;\bar{x}\right)\frac{e^{-x\beta (\bar{x})}}{q(\bar{x})}$$
and satisfies the normalizations
(98)$$\int_{0}^{\infty }f\left(x\right)dx=1,\;\int_{0}^{\infty }xf\left(x\right)dx=\bar{x}.$$
The log of the cluster function $w(x;\bar{x})$ is the functional derivative of the selection functional at the MPD:(99)$$\log w\left(x;\bar{x}\right)=\frac{\delta \log W[f]}{\delta f}$$
and the notation $w(x;\bar{x})$ indicates this function of *x* will generally depend on $\bar{x}$ as well since the functional derivative of non linear functionals depend on the function on which the derivative is evaluated. Since the microcanonical probability peaks sharply about the MPD (we are assuming a stable single-phase state) all ensemble averages revert to averages over the continuous MPD. The ensemble average kernel is then equal to the mean kernel within the MPD
(100)$${\langle K\rangle }_{M,N}\to \bar{K}\left(\bar{x}\right)=\int_{0}^{\infty }dx\int_{0}^{\infty }dyK\left(x,y\right)f\left(x\right)f\left(y\right).$$
The log of the intensive partition function, $\log\omega \left(\bar{x}\right)=\log\Omega_{M,N}/N$, satisfies
(101)$$\log\omega =\beta \bar{x}+\log q,$$
with
(102)$$\beta =\frac{d\log\omega }{d\bar{x}}.$$
These are the intensive forms of Equations (Equation 40) and (Equation 42), respectively. The partition function of aggregation is obtained from Equation (Equation 32) by expressing the summation over $\log{\langle K\rangle }_{g}$ and an integral over $\bar{K}\left(\bar{x}\right)$:(103)$$\log\omega =1+\log\bar{x}+\bar{x}\int_{0}^{\bar{x}}\log\bar{K}\left(y\right)\frac{dy}{y^{2}}.$$
The parameter β is obtained from Equation (Equation 102) and logq from (Equation 101): (104)$$\begin{array}{c} \beta =\frac{1}{\bar{x}}+\frac{\log\bar{K}\left(\bar{x}\right)}{\bar{x}}+\int_{0}^{\bar{x}}\log\bar{K}\left(y\right)\frac{dy}{y^{2}}, \end{array}$$(105)$$\begin{array}{c} q=\frac{\bar{x}}{\bar{K}\left(\bar{x}\right)}. \end{array}$$
From Equation (Equation 54) we obtain
(106)$$\frac{1}{\bar{x}}\int_{0}^{\infty }dx\int_{0}^{x}f\left(x\right)\frac{w\left(x-y;\bar{x}\right)w\left(y;\bar{x}\right)}{w(x;\bar{x})}K\left(x-y,y\right)=1,$$
which expresses a condition on $w(x;\bar{x})$.

Equations (Equation 97), (Equation 103)–(Equation 106) provide an equivalent mathematical description of Smoluchowski aggregation in the continuous limit. These are accompanied by the variational condition
(107)$$\log\omega \geq -\int_{0}^{\infty }p\left(x\right)\log\frac{p(x)}{w(x;\bar{x})}dx,$$
which is the continuous form of Equation (Equation 48) and is satisfied by all distributions p(x) with mean $\bar{x}$. The equality defines the solution to the Smoluchowski process, the MPD, f(x).

As a demonstration we apply these results to the constant kernel. The right-hand side of Equation (Equation 106) is zero and we obtain W[f]=1 at all times. It follows that $w(x;\bar{x})=1$. The parameters β and *q* are
(108)$$\begin{array}{c} \beta =1/\bar{x},\;q=\bar{x}, \end{array}$$
and the MPD becomes
(109)$$f\left(x\right)=\frac{e^{-x/\bar{x}}}{\bar{x}}.$$
This is the well-known solution of the constant kernel in the continuous domain. The partition function of the constant kernel is
(110)$$\log\omega =1+\log\bar{x},$$
and the inequality in Equation (Equation 107) becomes
(111)$$1+\log\bar{x}\geq -\int p\left(x\right)\log p\left(x\right)dx=H\left[p\right],$$
whose right-hand side is the Shannon entropy of distribution p(x). For fixed $\bar{x}$ it is maximized by the exponential distribution, whose entropy is $1+\log\bar{x}$: the inequality is indeed satisfied.

## 8. Summary

With the results obtained here we have made contact with several previous works in the literature. The mean distribution for the constant kernel in Equation (Equation 59) was given by Hendriks [26], who also obtained a recursion for the partition function similar to that in Equation (Equation 30). The combinatorial treatment of Hendriks has in fact several common elements to ours but is limited to the constant and sum kernels and lacks the thermodynamic element of this work. The recursion for the cluster weights in Equation (Equation 55) has appeared in various treatments of aggregation, both deterministic [2,27] and stochastic [15,17,26]. The mean distributions in the continuous limit for the mean and sum kernels and for the product kernel in the pre-gel region are well known results in the literature [2]. The instability of power-law kernels has been discussed by Ziff et al. [3] in the context of the Smoluchowski equation. These connections to prior literature serve to validate the theory presented here and demonstrate that the thermodynamic treatment provides a unified theory of aggregation that brings previously disconnected results under a single formalism, the *Smoluchowski ensemble*.

The Smoluchowski ensemble is a probability space of distributions that are feasible under the rules of binary aggregation. The structure of this space, that is, the connectivity of the graph in Figure 1, is solely determined by the condition that aggregation is a binary event; the probability measure over this space is determined by the rate expression prescribed by the aggregation model. In Smoluchowski aggregation the rate is directly proportional to the number of clusters that appear on the reactant side of the aggregation reaction and on the aggregation kernel. In the scaling limit the probability of distribution is sharply peaked around a single distribution of the ensemble, its most probable distribution (MPD). In this limit all ensemble averages reduce to averages of the MPD, a distribution that alone suffices to generate all properties of the ensemble. The Smoluchowski coagulation equation is the time evolution of the most probable distribution in the asymptotic limit.

The step that turns the Smoluchowski ensemble into a *thermodynamic* ensemble is Equation (Equation 18). It expresses the probability of distribution in terms of two special functionals, the multinomial coefficient n! and the selection functional W(n). This formulation introduces the partition function $\Omega_{M,N}$ as the central property of the ensemble to which al other properties are connected. The thermodynamic calculus, summarized by the equations in Table 2, is a mathematical consequence of the variational condition that defines the most probable distribution in Equation (Equation 39) as the solution to the constrained maximization of the probability P(n) in Equation (Equation 18). The constraints are given by Equations (Equation 3) that fix the zeroth and first order moments of the distribution. These constraints define a *microcanonical* ensemble of distributions with fixed mean $\bar{x}=M/N$.

The MPD obtained by the method of constrained maximization is stable, provided that the partition function is concave in its independent variables. In extensive terms, $\log\Omega_{M,N}$ must be concave in M,N; in intensive terms, $\log\omega (\bar{x})$ must be concave in $\bar{x}$. The two conditions are equivalent and define the stability criterion of the MPD. As in regular thermodynamics, when the stability criterion is violated the system experiences *phase splitting* and exists a mixture of two phases—mathematically, as a linear combination of two independent MPD’s. In aggregation these phases are the *sol phase*, which is represented by the MPD in Equation (Equation 39) and the *gel phase* (giant component), which in the scaling limit is represented by a delta function at *∞*. The splitting into a sol and gel phase is treated by the theory in a natural and rigorous manner.

Notably, entropy in this treatment plays no special role. The Shannon entropy of distribution is the log of the multinomial coefficient. In the scaling limit, entropy is a component of the partition function through Equation (Equation 36),
(112)$$\log\Omega_{M,N}=H\left(n^{*}\right)+\log W\left(n^{*}\right),$$
where $H(n^{*})$ is the Shannon functional evaluated at the MPD. In the special case of the constant kernel $W(n^{*})=1$. In this case the partition function reduces to the Shannon entropy of the MPD,
(113)$$\log\Omega_{M,N}=-N\sum_{i}\frac{n_{i}}{N}\log\frac{n_{i}}{N};\;(\mathrm{constant}\;\mathrm{kernel}),$$
and the variational condition reads,
(114)$$H\left(n\right)\leq H\left(n^{*}\right)=\Omega_{M,N};\;(\mathrm{constant}\;\mathrm{kernel}).$$
In this form we have recovered the inequality of the second law as stated in statistical thermodynamics: the entropy of the equilibrium distribution (MPD) is at maximum with respect to all feasible distributions, namely, all distributions with the same mean. As is well known this distribution is exponential. The constant kernel is special. With W(n)=1 the probability of distribution is proportional to n!; accordingly, all ordered sequences of *N* clusters with total mass *M* are equally probable. The ordered sequence of cluster masses in this case is analogous to microstate in statistical mechanics and the condition W=1 analogous to the postulate if equal a priori probabilities. In the general case the Shannon entropy and the log of the microcanonical partition function are not the same. The fundamental functional that is maximized is the microcanonical weight n!W(n), whose log is
(115)H(n)+logW(n).
The selection functional incorporates the effect of the aggregation kernel and in this sense it the point of contact between thermodynamics and the mathematical model of the stochastic process that gives rise to the probability space of interest. In Smoluchowski aggregation the model is defined by the transition rate in Equation (Equation 10) and the corresponding governing equation for *W* is Equation (Equation 33).

The thermodynamic formalism developed here is not limited to aggregation. Two alternative derivations that make no reference to stochastic process that gives rise to the probability space have been given in References [20,28]. As long as logW is a homogeneous functional with degree 1, the thermodynamic relationships follow as a direct consequence of the maximization of the microcanonical probability in Equation (Equation 18) under the constraints in Equation (Equation 3). The details of aggregation enter through Equations (Equation 32) and (Equation 33) that give the partition function and selection functional in terms of the aggregation kernel. The approach may be generalized to other processes including growth by monomer addition and breakup. These will be treated elsewhere.

## Figures and Tables

**Figure 1 entropy-22-01181-f001:**
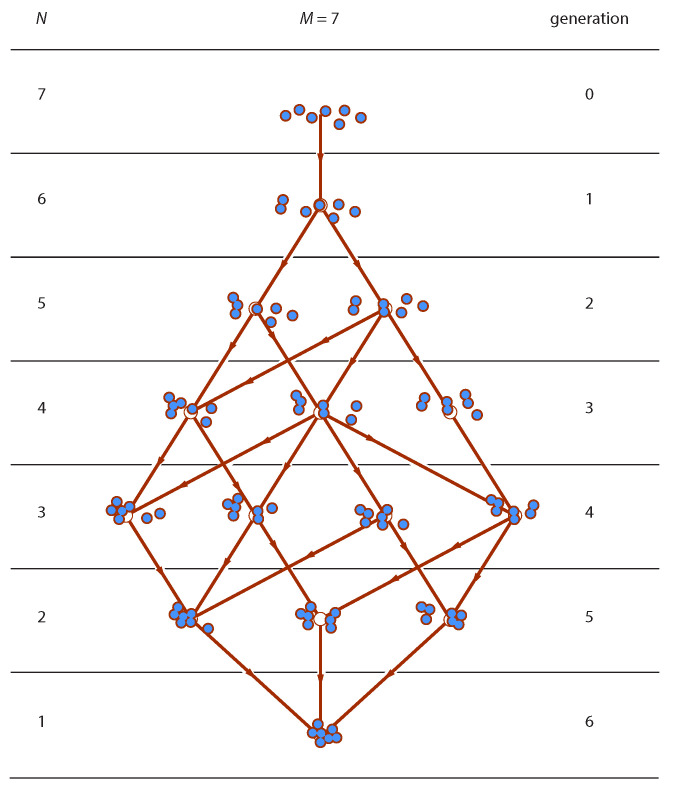
The aggregation graph for M=7. Each layer contains all feasible distributions in that generation.

**Figure 2 entropy-22-01181-f002:**
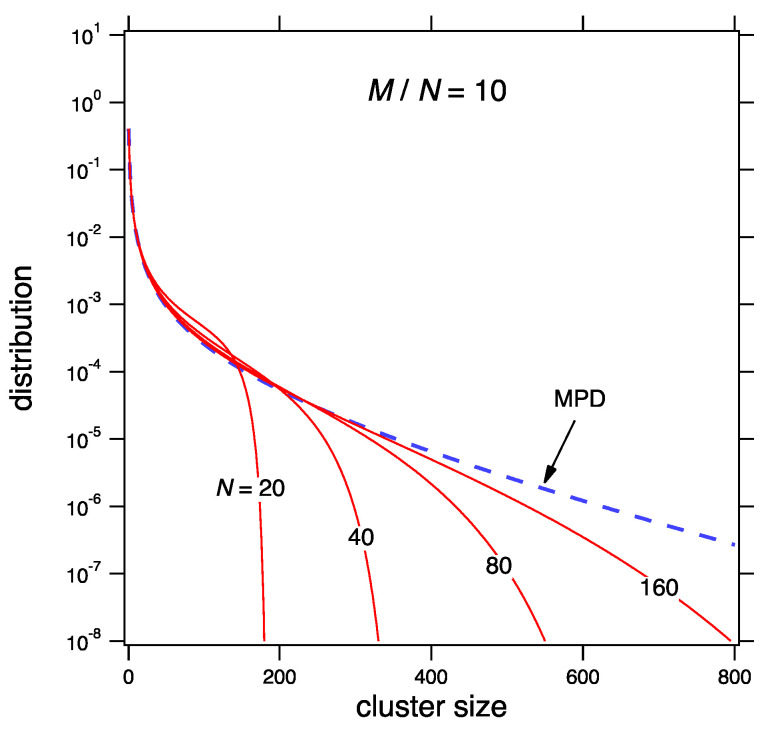
Approach to scaling limit for the sum kernel at $\bar{x}=10$(θ=0.9). The most probable distribution (MPD) is calculated from Equation (Equation 70) and the mean distribution from Equation (Equation 67) with $M=\bar{x}N$, N=20,40,80,160.

**Figure 3 entropy-22-01181-f003:**
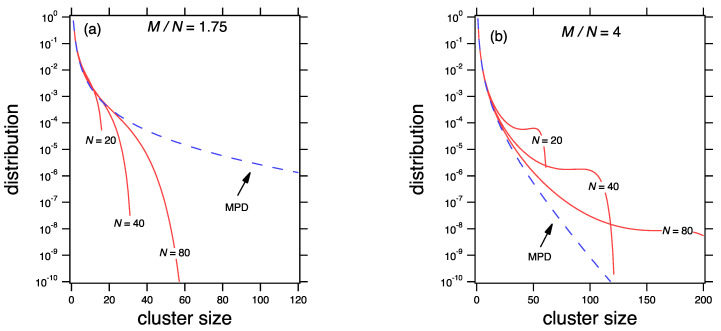
Approach to the scaling limit for the product kernel with (**a**) $\bar{x}=1.75$ and (**b**) $\bar{x}=4$. The MPD is calculated from Equation (Equation 71) and the mean distribution (dashed lines) from Equation (Equation 78). The distributions for $\bar{x}=4$ are not stable.

**Figure 4 entropy-22-01181-f004:**
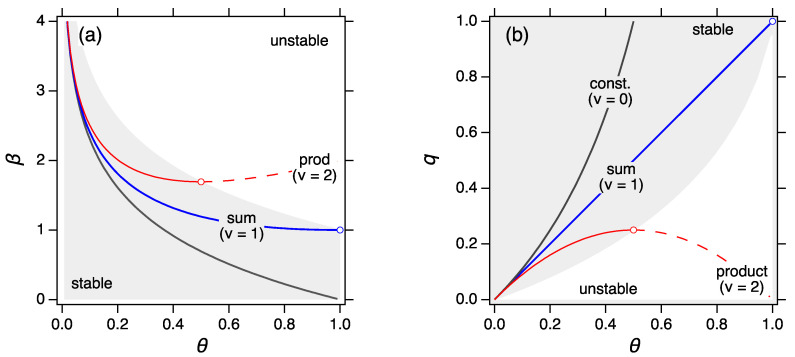
Phase diagram of power-law kernels: In the shaded region the system is stable and is represented by its MPD. The unshaded region is unstable and the system is split into two phases, a sol phase and a gel phase, each represented by its own MPD. (**a**,**b**) provide equivalent criteria of stability.

**Figure 5 entropy-22-01181-f005:**
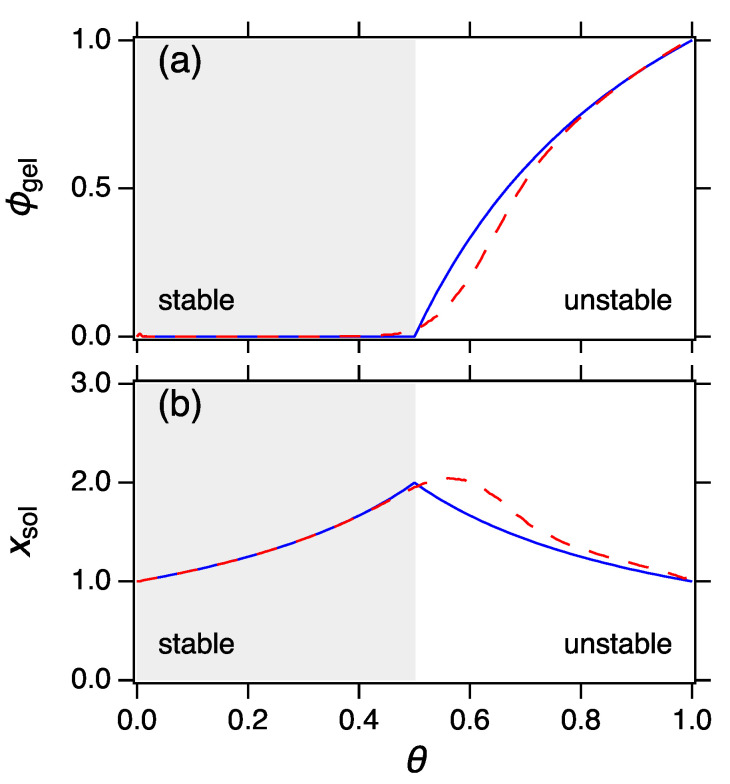
(**a**) Gel fraction and (**b**) mean sol cluster size as a function of the progress variable θ. Past the gel point the mean size in the sol retraces its pre-gel history back to its initial size ${\bar{x}}_{\mathrm{sol}}=1$. The dashed lines are Monte Carlo (MC) simulations with M=200 particles.

**Figure 6 entropy-22-01181-f006:**
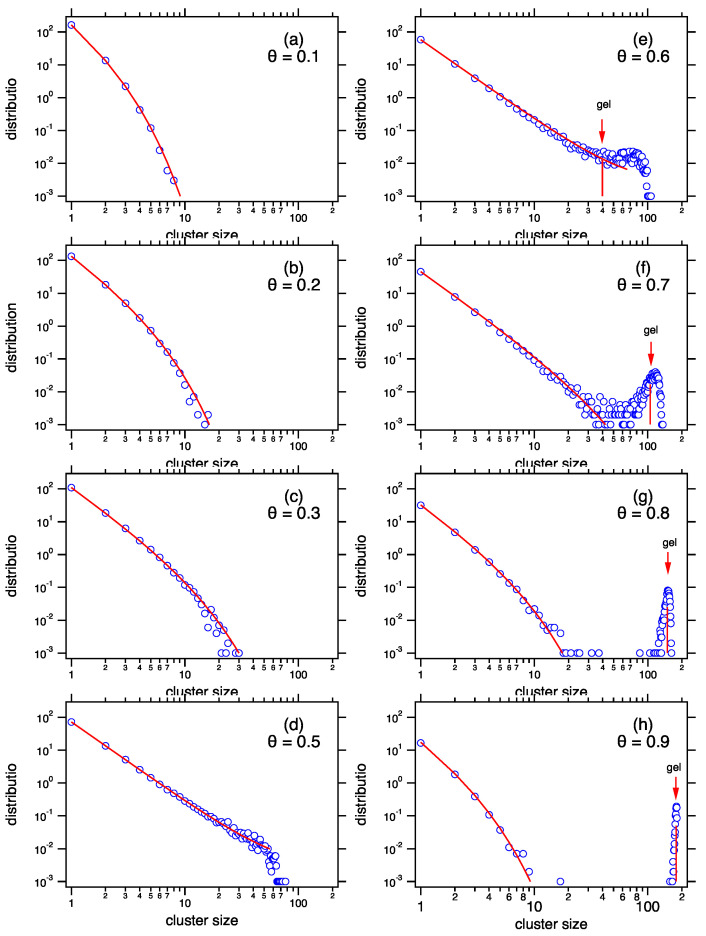
Monte Carlo snapshots of the mean distribution of the product kernel with M=200 particles (open circles are MC results, solid lines are calculated from theory). The gel phase emerges at $\theta^{*}=0.5$ and moves towards ever larger sizes (arrows mark the theoretical predictions). The distribution of the sol grows in the pre-gel region range 0<θ<0.5 but contracts once past the post-gel point (θ>0.5).

**Table 1 entropy-22-01181-t001:** Selected aggregation kernels.

Brownian coagulation	$K_{i,j}=\frac{1}{4}\left(2+{\left(\frac{i}{j}\right)}^{1/3}+{\left(\frac{j}{i}\right)}^{1/3}\right)$
Constant kernel	$K_{i,j}=1$
Flory/Stockmayer kernel	$K_{i,j}=\frac{\left(fi-2i+2)(fj-2j+2\right)}{f^{2}}$
Product kernel	$K_{i,j}=ij$
Sum kernel	$K_{i,j}=\frac{i+j}{2}$

**Table 2 entropy-22-01181-t002:** Summary of thermodynamic relationships.

Most Probable Distribution	$\frac{n_{k}^{*}}{N}=w_{k}^{*}\frac{e^{-\beta k}}{q}$	Equation (Equation 39)
Partition Function	$\Omega_{M,N}=\beta M+\left(\log q\right)N$	Equation (Equation 40)
	$\beta ={\left(\frac{\partial \log\Omega }{\partial M}\right)}_{N}$	Equation (Equation 42)
	$\log q={\left(\frac{\partial \log\Omega }{\partial M}\right)}_{M}$	Equation (Equation 43)
Gibbs-Duhem Equation	Mdβ+Ndlogq=0	Equation (Equation 44)
Variational Condition(Second Law)	$\frac{\log\Omega_{M,N}}{N}$ $\geq -\sum_{i}p_{i}\log\frac{p_{i}}{w_{i}^{*}}$	Equation (Equation 48)

**Table 3 entropy-22-01181-t003:** Summary of Constant, Sum and Product Kernel; in all cases $\theta =1-1/\bar{x}$.

	Constant Kernel	Sum Kernel	Product Kernel ${}^{†}$
$K_{i,j}$	1	(i+j)/2	ij
Ω	M−1N−1	N!MM−NM!M−1N−1	N!MM−NM!2M−1N−1
β	−logθ	θ−logθ	2θ−logθ
*q*	$\frac{\theta }{1-\theta }$	θ	θ(1−θ)
$w_{k}$	1	$\frac{k^{k-1}}{k!}$	$\frac{2^{k-1}k^{k-2}}{k!}$
MPD	$\left(1-\theta \right)\theta^{k-1}$	$\frac{k^{k-1}}{k!}\theta^{k-1}e^{k\theta }$	$\frac{{\left(2\theta k\right)}^{k-2}}{k!}\frac{2\theta }{1-\theta }e^{-2\theta k}$

${}^{†}$ Valid only for θ≤1/2.

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
