# Peer review of "The Smoluchowski Ensemble—Statistical Mechanics of Aggregation"

_entropy, 2020, doi:10.3390/e22101181_

Round 1

Reviewer 1 Report

This manuscript gives a very interesting presentation of the Statistical Mechanics of aggregation. The main document and related supplementary file are very well orgnized, with usefull demonstrations and results, and I recommend its publication.

I have the following comments/observations that I think is important to clarify:

1) It would be useful to have a more detailed discussion about the relation of the theory presented here and the seminal work by Stockmayer (1943) (doi.org/10.1063/1.1723803) In the present manuscript, it is not clear the specific conditions that the aggregation Kernel Ki,j needs to satisfy, whereas in the case of the Stockmayer model, this author is very precise about the molecular mechanims that justify the modeling given by the Smoluchowski equation. Since the aggregation theory developed in the manuscript intends to be more general and applicable to a wide range of systems, what are the conditions/restrictions for  Ki,j?

2) The approach given in the manuscript seems to be restricted to extensive entropic systems. The non-entropic cases can be incorporated within the same framework? An interesting issue could arise in the derivation given in the 3.2 section if the following two terms are used in the Stirling approximation. 

3) In physical terms, what would be the meaning of the "mean-field" approximation used in the derivation of Eq. 18? , in terms of the aggregation process?

4) Eq. (44) has a  mistake, "M" is repeated as the partial derivative after Eq. (43), M is fixed variable in (44).

5) there are minor grammatical mistakes in lines 128-129, 283-284

Reviewer 2 Report

Please see attached pdf.

Round 2

Reviewer 2 Report

I now propose the manuscript to be accepted for publication.